# Quality Estimation-Assisted Automatic Post-Editing

**Sourabh Deoghare[1], Diptesh Kanojia[1,2], Tharindu Ranasinghe[3],**
**Frédéric Blain[4] and Pushpak Bhattacharyya[1]**
[1]CFILT, Indian Institute of Technology Bombay, Mumbai, India
[2]Surrey Institute for People-Centred AI, University of Surrey, United Kingdom
[3]Aston University, Birmingham, United Kingdom
[4]Tilburg University, Tilburg, The Netherlands
{sourabhdeoghare, pb}@cse.iitb.ac.in, d.kanojia@surrey.ac.uk,
t.ranasinghe@aston.ac.uk, f.l.g.blain@tilburguniversity.edu

## Abstract

Automatic Post-Editing (APE) systems are prone to over-correction of the Machine Translation (MT) outputs. While a Word-level Quality Estimation (QE) system can provide a way to curtail the over-correction, a significant performance gain has not been observed thus far by utilizing existing APE and QE combination strategies. This paper proposes joint training of a model over QE (sentence- and word-level) and APE tasks to improve the APE. Our proposed approach utilizes a multi-task learning (MTL) methodology, which shows significant improvement while treating the tasks as a 'bargaining game' during training. Moreover, we investigate various existing combination strategies and show that our approach achieves state-of-the-art performance for a 'distant' language pair, *viz.,* English-Marathi. We observe an improvement of 1.09 TER and 1.37 BLEU points over a baseline QE-Unassisted APE system for English-Marathi while also observing 0.46 TER and 0.62 BLEU points improvement for English-German. Further, we discuss the results qualitatively and show how our approach helps reduce over-correction, thereby improving the APE performance. We also observe that the degree of integration between QE and APE directly correlates with the APE performance gain. We release our code publicly[1].

## 1 Introduction

Significant progress in Machine Translation (MT) and adopting MT systems in the translation industry has led to motivated research on auxiliary tasks. One auxiliary task in the MT field is Automatic Post-editing (APE), which is aimed toward the automatic identification and correction of translation errors. The APE task is motivated by the need to improve MT output in black-box conditions, where translation models are not accessible for modification or retraining (Chatterjee et al., 2020), which

can be costly in terms of time and effort. Quality Estimation (QE) is another auxiliary task that assesses the quality of MT output in the absence of reference translations (Specia et al., 2020). QE is performed at a word, sentence, and document level. A word-level QE system predicts an OK or BAD tag for each token on both source and target segments, while a sentence-level QE assigns a rating (0-100), referred to as a Direct Assessment (DA) score, to the translation denoting its overall quality.

Both QE and APE have been explored significantly as standalone tasks. Over the years, predicting MT quality has been formulated differently (*e.g.* ranking, scoring, qualitative metrics) and has been dealt with at diverse levels of granularity. Consistent advancements in both these tasks make QE and APE more appealing technologies. Despite the growing interest in the two tasks and their intuitive relatedness, previous research in both areas has mostly followed separate paths. Current QE integration strategies in the MT-APE pipeline have only been applied over *statistical phrase-based MT system translations* (Chatterjee et al., 2018).

Our **motivation** for the current work is: the potential usefulness of leveraging the two technologies to develop a better APE system has been scarcely explored. Also, there is no systematic analysis of APE performance when different QE and APE combination strategies are used to post-edit translations obtained from *a Neural Machine Translation (NMT) system*. We hypothesize that QE can help APE in the following manner: A sentence-level QE output can provide an overall idea of how much editing is required to correct a translation, whereas a word-level QE output can help APE by identifying translation tokens that need editing.

Our goal is to reduce over-correction in APE by using QE. Hence, our contributions are:

1. Joint training over QE (sentence- and word-level) and APE tasks, which helps APE learn to identify and correct erroneous translation

---

[1]https://github.com/cfiltnlp/APE_MTL

segments, leading to significant (based on the significance test with $p$ value of 0.05) performance gains for English-Marathi (1.09 TER) and English-German (0.46 TER) over the QE-unassisted APE baseline. (Refer Table 4)

2. A novel approach to multi-task learning (Nash-MTL) applied to QE and APE, which treats learning both tasks as a bargaining game. The improvements obtained over the baseline are given in the first point.

3. A comprehensive study investigating known QE and APE combination strategies while showing that a tighter coupling of both tasks is increasingly beneficial in improving APE performance. For this study, we consider three existing QE and APE combination strategies (QE as APE Activator, QE as MT/APE Selector, QE as APE Guide) and the proposed method (Joint Training over QE and APE). (Refer Table 5 or Figure 4)

## 2 Related Work

During the past decade, there has been tremendous progress in the field of QE and APE, primarily due to the shared tasks organized annually by the Conference on Machine Translation (WMT), since 2012 and 2015, respectively (Zerva et al., 2022; Bhattacharyya et al., 2022).

Läubli et al. (2013) and Pal et al. (2016) have shown that the APE systems have the potential to reduce human effort by correcting systematic and repetitive translation errors. Recent APE approaches utilize transfer learning by adapting pre-trained language or translation models to perform post-editing (Lopes et al., 2019; Wei et al., 2020; Sharma et al., 2021). Also, the recent approaches use multilingual or cross-lingual models to get latent representations of the source and target sentences (Lee et al., 2020). Oh et al. (2021) have shown that gradually adapting pre-trained models to APE by using the Curriculum Training Strategy (CTS) improves performance. Deoghare and Bhattacharyya (2022) have demonstrated that augmenting the APE data by phrase-level APE triplets improves feature diversity and used the CTS to train the APE system on high-quality data.

Recently, in the field of QE, neural-based systems such as deepQuest (Ive et al., 2018) and OpenKiwi (Kepler et al., 2019) have consistently outperformed other approaches in WMT Quality

Estimation shared tasks (Kepler et al., 2019). These systems utilize an encoder-decoder Recurrent Neural Network (RNN) architecture, commonly called the 'predictor,' combined with a bidirectional RNN known as the 'estimator,' which generates quality estimates. However, one drawback of these architectures is that they require extensive predictor pre-training, relying on large parallel data and demanding computational resources (Ive et al., 2018). To address this limitation, TransQuest (Ranasinghe et al., 2020b) emerged as a solution, winning the WMT20 sentence-level QE (DA prediction) shared task Specia et al. (2020). TransQuest eliminates the need for a predictor by leveraging cross-lingual embeddings. The authors fine-tuned an XLM-Roberta model for the sentence-level QE, demonstrating that a simple architecture can achieve state-of-the-art results. Subsequently, the TransQuest framework has been extended to the word-level QE task (Ranasinghe et al., 2021). Recently, Deoghare and Bhattacharyya (2022) showed that combining sentence-level and word-level QE systems can help alleviate the problem of inconsistent predictions.

Martins et al. (2017) used APE outputs to improve QE systems. Hokamp (2017) used an ensemble of factored NMT models for word-level QE and APE tasks. Chatterjee et al. (2018) compared three combination approaches and showed the potential of QE systems in improving APE on output obtained from a phrase-based MT system. We use these three approaches for comparison in the current work. The winning submission of the WMT22 APE shared task shows the usefulness of a sentence-level QE system in deciding whether APE has improved a translation or not (Bhattacharyya et al., 2022).

## 3 Standalone APE and QE Systems

This section discusses the approaches used to develop standalone APE and QE systems. We follow the state-of-the-art approaches for training the systems as it allows us to investigate whether the findings of Chatterjee et al. (2018) hold when neural APE and QE systems are used to post-edit and assess the quality of NMT-generated translations.

### 3.1 Automatic Post-Editing

The subsection describes a standalone neural APE system that we use as a baseline as well and refer to it as *APE w/o QE*.

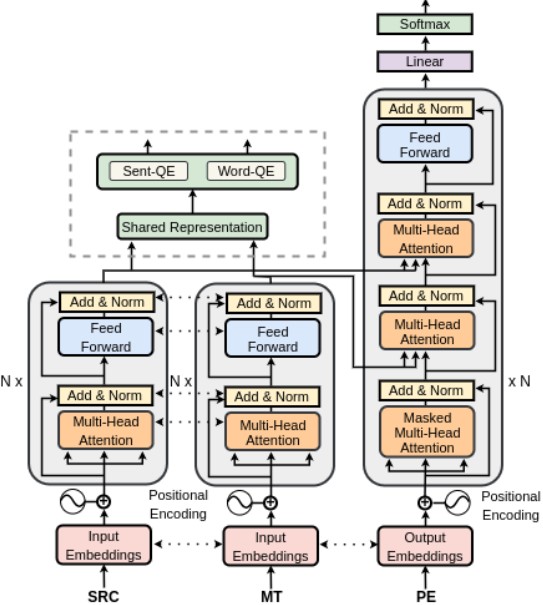

Figure 1: Dual-encoder single-decoder architecture with task-specific heads used for joint training over QE and APE tasks for the English-Marathi pair. Without components shown within the dashed rectangle, the diagram depicts the architecture of a standalone APE system. Dashed arrows represent tied parameters and common embedding matrices for the encoders and the decoder.

**Architecture**   We develop the APE system using transformer-based encoder-decoder architecture. We use two separate encoders for the English-Marathi APE system to encode a source sentence and its translation, as these languages do not share script or vocabulary. Outputs of both encoders are passed to two consecutive cross-attention layers in the decoder. An architecture shown in Figure 1 without the Sentence-QE and Word-QE heads represents the English-Marathi APE architecture. While for the English-German APE system, we use a single-encoder single-decoder architecture as there is a script and vocabulary overlap between these two languages. Therefore, a single encoder encodes the concatenation of source and translation generated by adding a '<SEP>' tag between them, and the encoder output is passed to a single cross-attention layer in the decoder. For both the pairs, the encoders are initialized using IndicBERT (Kakwani et al., 2020) weights.

**Dataset**   We use datasets released through the WMT21 (Akhbardeh et al., 2021) and WMT22 (Bhattacharyya et al., 2022) English-German and English-Marathi APE shared tasks, respectively, to conduct the experiments. The

APE data consists of real (human-generated post-edits) and synthetic (artificial post-edits) (Junczys-Dowmunt and Grundkiewicz, 2016). The English-Marathi APE dataset contains 18K real APE triplets from General, News, and Healthcare domains in the train set, and the synthetic APE data contains around 2.5M triplets from multiple domains. The train set for the English-German pair contains 7K real APE triplets from the general domain, and the synthetic APE data, eSCAPE (Negri et al., 2018), contains around 4M shuffled triplets from multiple domains. We use the corresponding development sets containing 1K APE triplets to evaluate the APE systems.

We also use the parallel corpora during the APE training phase. For the English-Marathi pair, we use Anuvaad[2], Samanantar (Ramesh et al., 2022), and ILCI (Bansal et al., 2013) datasets containing around 6M sentence pairs. While for the English-German pair, we use the News-Commentary-v16 WMT22 MT task dataset (Kocmi et al., 2022) of around 10M sentence pairs.

**Data Augmentation and Pre-processing**   We augment the synthetic APE data with automatically generated phrase-level APE triplets. First, we train source-to-translation and source-to-post-edit phrase-based statistical MT systems by employing Moses (Koehn et al., 2007). The phrase pairs from both MT systems are extracted in the next step. Then, we form the APE triplet by matching the source side of both phrase pairs. We control the quality of synthetic APE triplets (including the phrase-level APE triplets) by performing LaBSE-based filtering (Feng et al., 2022) and filter the low-quality triplets from the synthetic APE data. It is done by computing cosine similarity between the normalized embeddings of a source sentence and its corresponding post-edited translation. We retain the triplet if the cosine similarity is more than 0.91. We get around 50K and 60K phrase-level triplets for English-Marathi and English-German pairs, respectively.

**Model Training**   We follow the Curriculum Training Strategy (CTS) similar to Oh et al. (2021) for training our APE systems. It involves gradually adapting a model to more and more complex tasks. The steps of the CTS are described below.

In the first step, we train a single-encoder single-decoder model for performing source-to-target lan-

[2]Anuvaad Parallel Corpus

guage translation using the parallel corpus. In the next step, we add another encoder to the encoder-decoder model for English-Marathi APE while we use the same architecture for the English-German APE. We train the resulting model for the APE task using the synthetic APE data in the two phases for English-Marathi and in one phase for English-German. In the first phase, we train the model for the APE task using the out-of-domain APE triplets (*i.e.* any domains except the General, News, and Healthcare for English-Marathi). In the second phase, we train the model using the in-domain synthetic APE triplets. As the English-German APE data is of general (news or wiki) domain, we train the English-German APE model in a single phase using all synthetic data. Finally, we fine-tune the APE model using in-domain real APE data. Equation 1 shows the cross-entropy loss function used to train the APE model.

$$L_{APE} = -\sum_{w=1}^{|S|} \sum_{e=1}^{|V|} y_{w,e} \log\left(\hat{y}_{w,e}\right) \qquad (1)$$

Where $|S|$ and $|V|$ denote the number of tokens in the sentence and the number of tokens in the vocabulary, respectively. The APE output is denoted by the $\hat{y}_{w,e}$, while $y_{w,e}$ represents the ground truth.

### 3.2 Quality Estimation

This section describes the standalone sentence- (Sent-QE) and word-level (Word-QE) QE systems.

**Architecture** We use a transformer encoder to develop the QE models. To obtain representations of the input (concatenated source sentence and its translation), we employ XLM-R (Conneau et al., 2020). This model is trained on a massive multilingual dataset of 2.5TB, which includes 104 different languages, and the training is conducted using the masked language modeling (MLM) objective, similar to RoBERTa (Liu et al., 2019). The WMT20 sentence- and word-level QE shared task winning systems have utilized XLM-R-based QE models (Ranasinghe et al., 2020a; Lee, 2020). Therefore, we adopt a similar strategy for both QE tasks. We add a feedforward layer on the top of XLM-R to perform regression (sentence-level QE) and token-level classification (word-level QE).

**Dataset** For the QE tasks as well, we use datasets released through the WMT21 (Specia et al., 2021) and WMT22 (Zerva et al., 2022) QE shared tasks.

The WMT21 QE shared task data for English-German sentence- and word-level QE includes 7K and 1K instances in the train and development sets, respectively. For English-Marathi, the WMT22 dataset contains 26K and 1K instances in the train and development sets. For evaluating the QE systems, we use the corresponding development sets.

Each sample in the word-level English-German QE data consists of a source sentence, its translation, and a sequence of tags for tokens and gaps. The WMT22 dataset does not contain tags for gaps between translation tokens for the English-Marathi pair. So, we used the QE-corpus-builder[3] to obtain annotations for translations using their post-edits.

**Training Approach** We train XLM-R-based Sent-QE and Word-QE models for each language pair using the respective sentence- and word-level QE task datasets. During training, the weights of all layers of the model are updated.

**Sentence-level Quality Estimation Head** This task is modeled as a regression task. We use the hidden representation of the classification token (CLS) of the transformer model to predict normalized DA scores (Zerva et al., 2022) through the application of a linear transformation:

$$\hat{y}_{da} = W_{[CLS]}^T \cdot h_{[CLS]} + b_{[CLS]} \qquad (2)$$

where $\cdot$ denotes matrix multiplication, $W_{[CLS]} \in \mathcal{R}^{D \times 1}$, $b_{[CLS]} \in \mathcal{R}^{1 \times 1}$, and $D$ is the dimension of input layer $h$ (top-most layer of the transformer). Equation 3 shows the Mean Squared Error (MSE) loss used for this task.

$$\mathcal{L}_{sent} = MSE\left(y_{da}, \hat{y}_{da}\right) \qquad (3)$$

**Word-level Quality Estimation Head** We treat this task as a token-level classification task. We predict the word-level labels (OK/BAD) by applying a linear transformation (also followed by the softmax) over every input token from the last hidden layer of the XLM-R model:

$$\hat{y}_{word} = \sigma(W_{word}^T \cdot h_t + b_{word}) \qquad (4)$$

where $t$ marks which token the model is to label within a $T$-length window/token sequence, $W_{word} \in \mathcal{R}^{D \times 2}$, and $b_{word} \in \mathcal{R}^{1 \times 2}$. The cross-entropy loss utilized for training the model is depicted in Equation 5, which bears similarity to

---

[3] https://github.com/deep-spin/qe-corpus-builder

the architecture of MicroTransQuest as described in Ranasinghe et al. (2021).

$$\mathcal{L}_{word} = -\sum_{i=1}^{2} \Big( y_{word} \odot \log(\hat{y}_{word}) \Big)[i] \quad (5)$$

Refer Appendix A for the details about the hyper-parameters and the hardware used to conduct all the experiments.

## 4 Progressively Integrating QE with APE

This section discusses three existing strategies for combining QE with APE. Starting with lighter combination approaches, we move towards strongly coupled ones that help APE learn from the QE subtasks (sentence- and word-level QE).

### 4.1 QE as APE Activator

In this strategy, we use a QE system to decide whether a translation requires post-editing. We pass the source sentence and its translation to a sentence-level QE system to get a DA score prediction. If the DA score is below a decided threshold[4], only then do we use the APE system to generate the post-edited version of the translation.

### 4.2 QE as MT/APE Selector

Contrary to the *QE as APE Activator* approach, we use a QE system to decide whether an APE output is an improved version of a translation or not. This is done using a sentence-level QE system to get DA score predictions for the original translation and corresponding APE output. We consider the APE output as the final output only if it receives a higher DA score than the original translation.

### 4.3 QE as APE Guide

In the earlier two approaches, QE information is not passed to APE, but a sentence-level QE system is utilized merely to decide whether to APE or consider the APE output. Through the current approach, we explore a tighter combination of QE and APE by passing the sentence-level and word-level information obtained from the QE systems as additional inputs to APE. We use sentence-level QE systems to predict either DA scores or TER (Snover et al., 2006) scores. If we pass just the predicted DA or TER score to APE, we prepend it. If both scores are to be passed, then we modify the input format:

---
[4]decided empirically (Chatterjee et al., 2018).

<DA_Score > <TER_Score > <Source_sentence > <Target_sentence >. Similarly, to pass the word-level QE information, we add a <BAD > token before every source and translation token for which a BAD tag is predicted.

## 5 Joint Training over QE and APE

This approach investigates the tightest coupling between the QE tasks (sentence- and word-level QE) and APE. We follow exactly the same steps in the CTS (explained in Section 3.1), except the last one, and train the model for the APE task. During the last (fine-tuning) stage of the CTS, we jointly train the model on the APE and QE tasks using the real APE training data and the QE training data. To do so, we add the task-specific heads (refer Section 3.2) on top of a shared representation layer that receives inputs from the final encoder layers (Refer Figure 1). The representation layer has 2x neurons for En-Mr than for En-De. For En-De, its size is equal to the size of the final encoder layer. The following two multi-task learning (MTL) approaches are used to perform the experiments.

**Linear Scalarization (LS-MTL)** We use a straightforward MTL approach, LS-MTL, to combine the task-specific losses.

$$L_{LS-MTL} = L_{sent} + L_{word} + L_{APE} \quad (6)$$

All loss functions are weighed equally and are added together to get the combined loss ($L_{LS-MTL}$) as shown in Equation 6.

**Nash-MTL** To further explore the use of sophisticated MTL methods for training the APE model, we choose the Nash-MTL approach proposed by Navon et al. (2022), as the authors have shown that their proposed approach outperforms several other MTL methods through experiments on two different sets of tasks.

The utilization of MTL to jointly train a single model has been recognized as a means to reduce computation costs. However, conflicts arising from gradients of different tasks often lead to inferior performance of the jointly trained model compared to individual single-task models. To address this challenge, a widely used technique involves combining per-task gradients into a unified update direction using a heuristic. In this approach, the tasks negotiate for a joint direction of parameter update.

**Algorithm 1** Nash-MTL

---

**Input:** $\theta_0$ - initial parameter vector, $\{l_i\}_{i=1}^{K}$ - differentiable loss functions, $\eta$ - learning rate
**Output:** $\theta^T$
**for** $t = 1,..., T$ **do**
    Compute task gradients $g_i^t = \nabla_{\theta(t-1)} l_i$
    Set $G^{(t)}$ the matrix with columns $g_i^{(t)}$
    Solve for $\alpha : (G^t)^T(G^t)\alpha = 1/\alpha$ to obtain $\alpha^{(t)}$
    Update the parameters $\theta^{(t)} = \theta^{(t)} - \eta G^{(t)}\alpha^{(t)}$
**end for**
**return** $\theta^T$

---

The Nash-MTL approach considers a spherical region centered at the origin with a radius of $\epsilon$, for the MTL problem having parameters $\theta$. The update vectors are constrained within this sphere. The problem is formulated as a bargaining problem, where the center of the sphere represents the point of disagreement and the region serves as an agreement set. Each player's utility function is the dot product between the gradient vector $g_i$ of task $i$'s loss $l_i$ at $\theta$ and the update vector. Navon et al. (2022) showed that a Nash bargaining solution is a solution to $(G^t)^T(G^t)\alpha = 1/\alpha$. Algorithm 1 shows the process followed in the Nash-MTL method to update the parameters.

| | En-Mr | | En-De | |
|---|---|---|---|---|
| **Approach** | **TER** | **BLEU** | **TER** | **BLEU** |
| **Do-Nothing** | 22.93 | 64.51 | 19.06 | 68.79 |
| **APE w/o QE** | **19.39** | **68.35** | **18.91** | **68.91** |
| **QE –>APE** | 21.04 | 64.66 | 20.20 | 67.53 |

Table 1: Results on the English-Marathi (En-Mr) WMT22 and English-German (En-De) WMT21 development sets **when QE is used as an APE activator**.

## 6 Results and Analysis

This section discusses an evaluation of the QE and APE combination strategies. The performance of the APE systems is reported on the WMT development sets in terms of TER (Snover et al., 2006) and BLEU (Papineni et al., 2002) scores. We perform a statistical significance test considering primary metric (TER) using William's significance test (Graham, 2015).

For comparison, we use two APE **baselines** as follows: (i) a *Do-Nothing* baseline that does not modify original translations, as obtained from the

| | En-Mr | | En-De | |
|---|---|---|---|---|
| **Approach** | **TER** | **BLEU** | **TER** | **BLEU** |
| **Do-Nothing** | 22.93 | 64.51 | 19.06 | 68.79 |
| **APE w/o QE** | 19.39 | 68.35 | 18.91 | 68.91 |
| **APE –>QE** | **19.01** | **68.74** | **18.84** | **69.00** |

Table 2: Results on the English-Marathi WMT22 and English-German WMT21 development sets **when QE is used as MT/APE Selector**.

| | En-Mr | | En-De | |
|---|---|---|---|---|
| **Approach** | **TER** | **BLEU** | **TER** | **BLEU** |
| **Do-Nothing** | 22.93 | 64.51 | 19.06 | 68.79 |
| **APE w/o QE** | 19.39 | 68.35 | 18.91 | 68.91 |
| **+ DA** | 19.32 | 68.41 | 19.00 | 69.02 |
| **+ HTER** | 19.40 | 68.38 | 19.03 | 69.05 |
| **+ DA + HTER** | 19.35 | 68.47 | 19.01 | 69.01 |
| **+ BAD_TAGS** | **18.81** | **69.12** | 18.86 | 68.98 |
| **+DA + BAD_TAGS** | 18.86 | 69.05 | **18.84** | **69.02** |

Table 3: Results on the English-Marathi WMT22 and English-German WMT21 development sets **when QE model outputs are passed as additional inputs to APE**. **+DA**, **+HTER**: sentence-level QE (DA) model predictions and sentence-level QE (HTER) predictions passed as additional inputs to APE, respectively; **+BAD_TAGS**: APE input tokens annotated using the word-level QE predictions.

MT system. (ii) A standalone APE system that is unassisted by a QE system (*APE w/o QE*).

Experiments for the first two combination strategies *(QE as APE activator and QE as MT/APE selector)* are performed using the standalone APE and QE systems. For the *QE as APE Activator* experiments, we experiment with different thresholds of the DA scores and report the best results. The *QE as APE Guide* experiments use the same standalone sentence-level and word-level QE systems, but we modify the APE inputs from the second step of the CTS onwards, for each experiment, and train different APE models following the same training approach described in Section 3.

In the MTL-based experiments, first, we use the LS-MTL approach to perform various experiments using a combination of tasks to see the improvements brought by a simple MTL approach. Further, we try to optimize the performance on the best combination of tasks through the Nash-MTL approach.

**QE as APE Activator** We experimented with various threshold DA scores for each language pair and tried to find the optimal threshold value over a held-out subset taken from the train set. Thresholds of 0.32 and 0.35 DA scores give the best results

for English-Marathi and English-German language pairs, respectively.

The TER and BLEU scores obtained using this approach for both language pairs are reported in Table 1. We see a drop in performance compared to the APE w/o QE baseline system for both language pairs. For English-German, we observe that the performance is even poorer than the Do-Nothing baseline, which suggests the possibility that the APE can correct minor errors even in the translations that receive a very high DA score. A possible reason for not seeing improvements over the standalone APE system could be that the sentence-level QE information is too coarse for APE to figure out whether a translation requires editing.

**QE as MT/APE Selector** Table 2 shows the helpfulness of this combination strategy. *We see 0.38 and 0.07 TER point improvements over the APE w/o QE baseline system for English-Marathi and English-German pairs, respectively.* The results highlight the problem of *over-correction* that the neural APE systems face. We conjecture that the sentence-level QE information may be too coarse to decide whether a translation needs post-editing. Still, it can be used to compare the translations.

**QE as APE Guide** Table 3 reports the results of experiments when different combinations of the QE information are embedded into the APE inputs. It shows that passing sentence-level QE information to APE is not much effective. A possible reason could be the coarse nature of the sentence-level QE as discussed earlier. In particular, we see better performance when predicted DA scores are passed than the predicted HTER scores, which do not consider the semantic meaning.

We observe the best results when the source and translation tokens, tagged as BAD by the word-level QE, are annotated and passed to the APE for the English-Marathi pair. For the English-German pair, we get the best results when both the predicted DA scores and the word-level QE predicted information is passed as additional inputs to APE. *With these combinations, we get the 0.58 and 0.07 TER point improvements for the English-Marathi and English-German pairs, respectively.* It suggests that granular word-level QE information helps APE identify poorly translated segments, thereby helping it focus on their correction.

**Joint Training over QE and APE** Through these experiments, we explore the most robust cou-

pling of QE and APE. Table 4 compiles MTL-based results. We jointly train a model using LS-MTL in each experiment on a different combination of the APE and QE tasks. We observe better improvements by training on the word-level QE task and APE than when combining the sentence-level QE (DA prediction) task and APE. Further addition of the sentence-level QE (HTER prediction) does not significantly improve the performance. Unlike the findings from the *QE as APE Guide* experimental results, we get better improvements when sentence-level QE (DA prediction) and word-level QE tasks are used along with APE.

As jointly training a model using LS-MTL on the APE, Sent-QE (DA), and Word-QE tasks yields high improvements, we try to improve it using an advanced MTL method. Using the Nash-MTL method, we get further APE performance enhancements of 0.24 and 0.15 TER points for English-Marathi and English-German pairs, respectively. *Similarly, gains over the APE w/o QE baseline system are 1.09 and 0.46 TER points.*

Additional analysis, like the efficiency of these combination strategies, the number of translations improved or deteriorated by the jointly trained systems, and types of editing performed by the models, is presented in Appendix B and Appendix C.

**Key Observation** Our quantitative analysis from the comparison between the improvements (Table 5, Figure 4) brought by each combination strategy over the APE w/o QE baseline shows that as the QE and APE coupling gets stronger, we see better enhancements in the APE performance.

**Qualitative Analysis** We compared the English-Marathi QE-assisted APE outputs (of the Nash-MTL-based method) with the corresponding APE w/o QE outputs. Our finding that QE helps APE mitigate the over-correction issue is supported by multiple examples from the sample of the data we analyzed. We show two examples in Figure 2.

The first example shows a fluent and highly adequate translation. Unlike the QE-unassisted APE system that modifies the translation and replaces three words, compromising adequacy, the QE-assisted APE leaves all translation words untouched, except the *'ghaaloon'*, which is tagged as BAD by the Word-QE, is changed to *'ghaatalelee'* which improves the fluency. In the second example, the APE w/o QE system has compromised the fluency by dropping the pronoun *'he'* in the trans-

| Approach | En-Mr | | En-De | |
|---|---|---|---|---|
| | TER | BLEU | TER | BLEU |
| **Do-Nothing** | 22.93 | 64.51 | 19.06 | 68.79 |
| **APE w/o QE** | 19.39 | 68.35 | 18.91 | 68.91 |
| **LS-MTL [APE, Word-QE]** | 18.78 | 69.18 | 18.62 | 69.28 |
| **LS-MTL [APE, Sent-QE (DA)]** | 19.52 | 68.92 | 18.89 | 68.90 |
| **LS-MTL [APE, Sent-QE (HTER)]** | 19.69 | 67.66 | 18.95 | 68.94 |
| **LS-MTL [APE, Word-QE, Sent-QE (DA)]** | 18.54 | 69.45 | 18.56 | 69.37 |
| **LS-MTL [APE, Word-QE, Sent-QE (DA), Sent-QE (HTER)]** | 18.54 | 69.44 | 18.60 | 69.31 |
| **Nash-MTL [APE, Word-QE, Sent-QE (DA)]** | **18.30** | **69.72** | **18.45** | **69.53** |

Table 4: Results on the English-Marathi WMT22 and English-German WMT21 development sets when a model is **jointly trained on the APE and QE tasks**.

| | Example 1 | Example 2 |
|---|---|---|
| **Source** | Description: The stone sculpture of Vishnu's head, wearing a high Kiritamukuta. | When the instrument is moved rapidly, the balls hit the stretched skin, and make the rhythmic sound. |
| **MT Output** | वर्णनः विष्णूच्या डोक्याची दगडी मूर्ती, उच्च कीर्तिमुकुट घालून. | जेव्हा हे यंत्र वेगाने हलवले जाते, तेव्हा चेंडू पसरलेल्या त्वचेवर आदळतात आणि लयबद्ध आवाज करतात. |
| **Reference** | वर्णनः विष्णूच्या डोक्याची दगडी मूर्ती, उच्च कीर्तिमुकुट घातलेली. | जेव्हा हे वाद्य वेगाने हलवले जाते, तेव्हा चेंडू पसरलेल्या चामड्यावर आदळतात आणि लयबद्ध आवाज करतात. |
| **APE w/o QE** | वर्णनः विष्णूच्या डोक्याची दगडी मूर्ती, उंच कीर्तिमुकुट घालत असलेली. | जेव्हा यंत्र वेगाने हलवले जाते, तेव्हा चेंडू ओढलेल्या चामड्यावर आपटतात आणि लयबद्ध आवाजाची निर्मिती करतात. |
| **Sent-QE** | 0.28 | 0.11 |
| **Word-QE** | वर्णनः विष्णूच्या डोक्याची दगडी मूर्ती, उच्च कीर्तिमुकुट घालून. | जेव्हा हे यंत्र वेगाने हलवले जाते, तेव्हा चेंडू पसरलेल्या त्वचेवर आदळतात आणि लयबद्ध आवाज करतात. |
| **APE with QE** | वर्णनः विष्णूच्या डोक्याची दगडी मूर्ती, उच्च कीर्तिमुकुट घातलेली. | जेव्हा हे यंत्र वेगाने हलवले जाते, तेव्हा चेंडू ओढलेल्या चामड्यावर आदळतात आणि लयबद्ध आवाज करतात. |

Figure 2: Post-edits from English-Marathi APE w/o QE and Joint training approach-based APE (APE with QE) systems (Refer Figure 3 for gloss and English transliterations of the Marathi sentences). Underlined words show differences between the MT Output and the APE reference. Incorrectly translated words are colored in red. Words/space highlighted with green and red represents the predicted OK and BAD tags, respectively. The Sent-QE row shows the predicted normalized DA scores. The Sent-QE and Word-QE predictions are from the Nash-MTL-based jointly trained model.

| Approach | En-Mr | | En-De | |
|---|---|---|---|---|
| | TER | BLEU | TER | BLEU |
| **Do-Nothing** | 22.93 | 64.51 | 19.06 | 68.79 |
| **APE w/o QE** | 19.39 | 68.35 | 18.91 | 68.91 |
| **QE as APE Activator** | 21.04 | 64.66 | 20.20 | 67.53 |
| **APE as MT/APE Selector** | 19.01 | 68.74 | 18.84[*] | 69.00 |
| **QE as APE Guide** | 18.81 | 69.12 | 18.84[*] | 69.02 |
| **Joint Training** | **18.30** | **69.72** | **18.45** | **69.53** |

Table 5: **Overall results** on the English-Marathi WMT22 and English-German WMT21 dev sets **using different QE and APE integration strategies**. [[*] indicates the improvement is not significant.]

lation and by translating the English phrase *'make the rhythmic sound'* literally. For this translation, the Sent-QE prediction suggests a need for some editing. The Word-QE correctly tags the word *'pasaralelyaa'* as 'BAD.' It wrongly tags the word *'yantra'* as 'OK' and the gap before the last word (suggesting missing words before it) as 'BAD.' We

see the QE-assisted APE corrects the translation of the BAD-tagged word but is unaffected by the wrongly tagged gap.

# 7 Conclusion and Future Work

In this paper, we have investigated four strategies to integrate QE systems into the MT-APE pipeline, intending to improve APE performance when translations to be post-edited are obtained from an NMT system. Experimental results highlight the complementary nature of APE and QE tasks, which can be best utilized through our proposed approach: joint training of a model that learns both tasks simultaneously. The quantitative results and our qualitative analysis suggest that the QE helps APE to address the over-correction problem.

Among the existing combination strategies, the QE as MT/APE Selector improves performance over QE-unassisted APE by discarding low-quality

APE outputs. APE Guide is a more beneficial approach as we observe even better improvements using the tighter combination strategy. It provides APE with explicit information about the overall translation quality and erroneous segments. Moreover, a comparison of these combination strategies shows that tighter coupling between QE and APE is increasingly beneficial for improving APE.

We plan to conduct a thorough qualitative analysis of QE-assisted APE systems to gain a deeper understanding of the contribution of QE. We also aim to strengthen the coupling between APE and QE to enhance APE performance further. Additionally, we intend to conduct a study to evaluate the potential of APE in improving QE systems.

## 8 Limitations

In the current work, we focus on using QE to improve APE, and we see that the joint training on the QE tasks and APE helps improve APE performance. This joint training may also benefit QE. However, we have not investigated it in the current work. We observe more minor performance improvements in the case of the English-German pair, having a strict Do-Nothing baseline, compared to the English-Marathi. It suggests that the law of diminishing returns affects the QE and APE combination strategies. Improvements obtained by the QE and APE coupling tend to get lower as the quality of original translations keeps improving. The current work has compared four QE and APE combination strategies using only two language pairs due to the unavailability of an adequate amount of good-quality APE and QE datasets for other language pairs. Current findings may be domain-specific, and future research could evaluate the proposed combination strategies on diverse domains to assess their generalizability and robustness.

## 9 Ethics Statement

Our APE and QE models are trained on publicly available datasets referenced in this paper. These datasets have been previously collected and annotated; no new data collection has been carried out as part of this work. Furthermore, these are standard benchmarks released in recent WMT shared tasks. No user information was present in the datasets, protecting the privacy and identity of users. We understand that every dataset is subject to intrinsic bias and that computational models will inevitably learn biased information from any dataset.

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

# A Training Details

To maintain uniformity across all the experiments, we use an identical set of settings for all the QE and APE experiments. Our APE models are trained using a batch size of 32. We specified a maximum of 1000 epochs for training, implementing early stopping with patience of 5. The Adam optimizer was employed with a learning rate of 5 x $10^{-5}$,

| | Example 1 | |
|---|---|---|
| **Source** | Description: The stone sculpture of Vishnu's head, wearing a high Kiritamukuta. | - |
| **MT Output** | वर्णनः विष्णूच्या डोक्याची दगडी मूर्ती, उंच कीर्तिमुकुट घालून. | Varnan (*Description*): Vishnoochya (*Vishnu's*) dokyaachee (*of head*) dagadee (*of stone*) moortee (*sculpture*), uchcha (*high*) keertimukut (*Kirtimukuta*) ghaaloon (*after wearing*). |
| **Reference** | वर्णनः विष्णूच्या डोक्याची दगडी मूर्ती, उंच कीर्तिमुकुट घातलेली. | Varnan: Vishnoochya dokyaachee dagadee moortee, uchcha keertimukut ghaatalelee (*worn*). |
| **APE w/o QE** | वर्णनः विष्णूच्या डोक्याची दगडी मूर्ती, उंच कीर्तिमुकुट घालत असलेली. | Varnan: Vishnoochya dokyaachee dagadee moortee, uncha (*unch*) keertimukut ghaalat (*being worn*) asalelee (*being done*). |
| **Sent-QE** | 0.28 | - |
| **Word-QE** | वर्णनः विष्णूच्या डोक्याची दगडी मूर्ती, उंच कीर्तिमुकुट घालून. | Varnan: Vishnoochya dokyaachee dagadee moortee, uchcha keertimukut ghaaloon. |
| **APE with QE** | वर्णनः विष्णूच्या डोक्याची दगडी मूर्ती, उंच कीर्तिमुकुट घातलेली. | Varnan: Vishnoochya dokyaachee dagadee moortee, uchcha keertimukut ghaatalelee. |
| | **Example 2** | |
| **Source** | When the instrument is moved rapidly, the balls hit the stretched skin, and make the rhythmic sound. | - |
| **MT Output** | जेव्हा हे यंत्र वेगाने हलवले जाते, तेव्हा चेंडू पसरलेल्या त्वचेवर आदळतात आणि लयबद्ध आवाज करतात. | Jevhaa (*when*) he (*that*) yantra (*machine*) vegaane (*with speed*) halawale (*is moved*) jate (*done*), tevhaa (*at that time*) chendu (*balls*) pasaralelya (*spread*) twachewar (*on skin*) aadalataat (*hit*) aani (*and*) layabaddha (*rhythmic*) aavaaj (*sound*) karataat (*make*). |
| **Reference** | जेव्हा हे वाद्य वेगाने हलवले जाते, तेव्हा चेंडू पसरलेल्या चामड्यावर आदळतात आणि लयबद्ध आवाज करतात. | Jevhaa he vadya (*instrument*) vegaane halawale jate, tevhaa chendu pasaralelya chaamadyaawar (*on animal skin*) aapataataat aani layabaddha aavaaj karataat. |
| **APE w/o QE** | जेव्हा यंत्र वेगाने हलवले जाते, तेव्हा चेंडू ओढलेल्या चामड्यावर आपटतात आणि लयबद्ध आवाजाची निर्मिती करतात. | Jevhaa yantra (*machine*) vegaane halawale jaate, tevhaa chengdu odhalelyaa chaamadyaawar aapatataat (*hit*) aani layabaddha aavaajaachi (*of sound*) nirmitee (*to create*) karataat. |
| **Sent-QE** | 0.11 | - |
| **Word-QE** | जेव्हा हे यंत्र वेगाने हलवले जाते, तेव्हा चेंडू पसरलेल्या त्वचेवर आदळतात आणि लयबद्ध आवाज करतात. | Jevhaa he yantra vegaane halawale jate, tevhaa chendu pasaralelya twachewar aadalataat aani layabaddha aavaaj karataat. |
| **APE with QE** | जेव्हा हे यंत्र वेगाने हलवले जाते, तेव्हा चेंडू ओढलेल्या चामड्यावर आदळतात आणि लयबद्ध आवाज करतात. | Jevhaa he yantra vegaane halawale jaate, tevhaa chengdu odhalelyaa chaamadyaawar aadalataat aani layabaddha aavaaj karataat. |

Figure 3: Post-edits from English-Marathi APE w/o QE and Joint training approach-based APE (APE with QE) systems. The third column contains the gloss (shown in the brackets) and transliterations of Marathi sentences. Underlined words show differences between the MT Output and the APE reference. Incorrectly translated words are colored in red. Words/space highlighted with green and red represents the predicted OK and BAD tags, respectively. The Sent-QE row shows the predicted normalized DA scores. The Sent-QE and Word-QE predictions are from the Nash-MTL-based jointly trained model.

along with $\beta_1$ set to 0.9, and $\beta_2$ set to 0.997. Additionally, we utilized 25,000 warmup steps. On the decoder side, beam search was applied with a beam size of 5. For the QE experiments, we use a batch size of 16. We start with a learning rate of $2e-5$ and use $5\%$ of training data for warm-up. We use early stopping and patience over 20 steps. In the QE experiments and all MTL-based experiments, we used WandB for hyperparameter search. All the experiments are performed using NVIDIA A100 GPUS. The APE model contains around 40M parameters and training the model using CTS takes about 48 hours. The QE model has about 125M parameters and training one QE model takes around

2.25 hours. For pre-processing the English and German data, we used the NLTK library[5], and the IndicNLP library[6] is used for processing Marathi text. We used Pytorch[7] for Model training and inference. For computing the TER scores, we use the official WMT APE and QE evaluation script[8], and for computing the BLEU scores, we use the SacreBLEU[9] library.

---

[5]https://www.nltk.org/
[6]https://github.com/anoopkunchukuttan/indic_nlp_library
[7]https://pytorch.org/
[8]https://github.com/sheffieldnlp/qe-eval-scripts
[9]https://github.com/mjpost/sacrebleu

| Approach | Time (in Seconds) |
|---|---|
| **APE w/o QE** | **55.70** |
| **QE as APE Activator** | 66.18 |
| **QE as MT/APE Selector** | 109.07 |
| **Joint Training** | 73.96 |

Table 6: Inference time on 1K English-Marathi samples of the WMT22 dev set.

| | En-Mr (%) | En-De (%) |
|---|---|---|
| **Modified** | 43.8 | 46.2 |
| **Improved** | 65.5 | 45.6 |
| **Deteriorated** | 26.8 | 38.9 |
| **Precision** | 71.1 | 54.0 |

Table 7: Percentage (%) of translations in the WMT22 English-Marathi and WMT21 English-German dev sets modified, improved, and deteriorated by the APE-QE jointly trained model. The last row reports the precision, computed as a ratio of the number of translations improved to the total number of translations modified.

| Edit Operation | En-Mr (%) | En-De (%) |
|---|---|---|
| **Insertion** | 33.8 | 18.5 |
| **Deletion** | 52.3 | 63.0 |
| **Substitution** | 7.1 | 8.25 |
| **Shifting** | 6.9 | 10.2 |

Table 8: Statistics showing the percentage of insertion, deletion, substitution, and shifting operations performed by the APE-QE jointly trained models on WMT22 English-Marathi and WMT21 English-German dev sets.

## B  Efficiency of APE Systems

Table 6 shows the inference time of the four APE-QE coupling strategies for 1K Marathi translations on the same single GPU. The first two QE-APE combination strategies are notably faster than the latter two. The 'QE as APE Guide' strategy shows the highest latency, likely due to word-level tag generation from QE and then longer inputs passed to APE. Even though it is pipeline-free, the fourth strategy exhibits somewhat higher latency than the second strategy; the likely reason could be that it involves word-level tag generation. To summarize, strategy choice has a high impact on efficiency. Despite having slightly higher latency than the 'QE as APE Selector' strategy, the 'QE-APE MTL-based approach' remains the optimal choice due to its superior overall performance.

## C  Additional Quantitative Analysis

We analyzed the number of translations improved and deteriorated by the jointly trained APE models on the QE and APE tasks. The results are reported in Table 7. Also, the number of edit operations performed by these jointly trained models are compiled in Table 8

## D  Example Post-Edits with English Transliteration

Figure 3 contains English transliterations of the Marathi sentences shown in Figure 2.

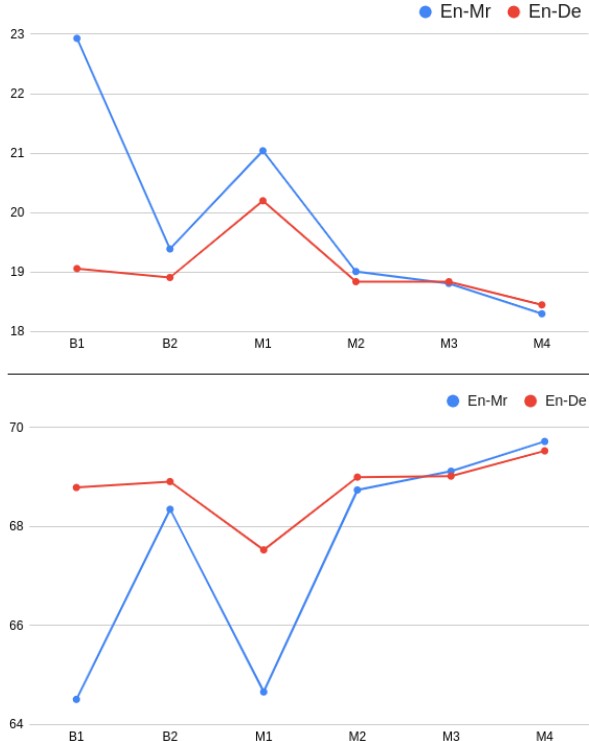

Figure 4: TER (top) and BLEU (bottom) scores for both language pairs **where a progression can be seen as we start coupling QE and APE tightly**. B1, B2: Do-Nothing and QE-Unassisted APE baselines; M1: QE as APE Activator; M2: QE as MT/APE Selector; M3: QE as APE Guide; M4: Joint Training.