# OpenReview forum: "Quality Estimation-Assisted Automatic Post-Editing"
_EMNLP/2023/Conference — EMNLP 2023 Findings_

### Official Review · Reviewer_aQPt · 2023-08-02

**Soundness:** 4

**Excitement:**

3: Ambivalent: It has merits (e.g., it reports state-of-the-art results, the idea is nice), but there are key weaknesses (e.g., it describes incremental work), and it can significantly benefit from another round of revision. However, I won't object to accepting it if my co-reviewers champion it.

**Paper Topic And Main Contributions:**

This paper addresses the issue of over-correction by Automatic Post-Editing (APE) systems when processing Machine Translation (MT) outputs. In particular, it investigates how Word-level Quality Estimation (QE) systems could be used to limit over-correction, an area where little performance gain has previously been observed. They carry out several experiments to explore the nature of combining APE and QE, and how it affects translation quality. They show that a framework of joint training with Nash - Multitask Learning performs the best, and obtains significant improvements in English-to-Marathi and English-to-German language directions.

**Questions For The Authors:**

A. Do you have an indication of how different strategies of integrating the two tasks impact efficiency?

B. It was not entirely clear to me which parameters are shared. Figure 1 shows that all of the encoder parameters are tied for both languages, but the caption also mentions the decoder parameters. Could you elaborate?

**Reasons To Accept:**

The paper is well written and easy to follow. The authors do a great job at introducing the two tasks and motivating why they could benefit from each other. They investigate several ways of combining the tasks, and carry out multiple ablation studies for each one of them. Their results in both tested language directions show significant improvements by using the proposed methodology.

**Reasons To Reject:**

I do not see any compelling reason for concern.

**Reproducibility:**

4: Could mostly reproduce the results, but there may be some variation because of sample variance or minor variations in their interpretation of the protocol or method.

**Reviewer Confidence:**

3: Pretty sure, but there's a chance I missed something. Although I have a good feel for this area in general, I did not carefully check the paper's details, e.g., the math, experimental design, or novelty.

---

> ### Author Rebuttal · Authors · 2023-08-28
>
> Thank you for reviewing our work and providing valuable comments and suggestions!
>
> ****
>
> ### Question A:
> - We interpret this question as how the QE-APE combination strategy impacts the post-edit generation time.
>
> - We measured the post-edit generation time of each of the four strategies for 1K Marathi translations on the same single GPU:
>     1. QE as APE Activator: 53.70 seconds
>     2. QE as APE Selector: 66.18 seconds
>     3. QE as APE Guide: 109.07 seconds
>     4. QE-APE MTL-based Approach: 73.96 seconds.
>
> - The first two QE-APE combination strategies were notably faster than the latter two. The ‘QE as APE Guide’ strategy showed the highest latency, likely due to word-level tag generation from QE and then longer inputs passed to APE. Even though it is pipeline-free, the fourth strategy exhibits somewhat higher latency than the second strategy; again, the likely reason could be that it involves word-level tag generation.
> - To summarize, strategy choice had a high impact on efficiency. Despite having slightly higher latency than the ‘QE as APE Selector’ strategy, the ‘QE-APE MTL-based approach’ remains the optimal choice due to its superior overall performance.
> - We will include these statistics and discussion in the appendix.
>
> ### Question B:
> It is our mistake that the horizontal dotted arrow is not present between the encoder and the decoder in the figure. The caption is correct. We have noted it, and we will update the figure. Thank you for pointing it out.

---

### Official Review · Reviewer_WKNZ · 2023-08-04

**Soundness:** 4

**Excitement:**

4: Strong: This paper deepens the understanding of some phenomenon or lowers the barriers to an existing research direction.

**Paper Topic And Main Contributions:**

The paper proposes joint training of a model over QE (at sentence-level and at word-level) and APE tasks to improve Automatic Post Editing. The authors apply a multi-task learning strategy.

They present a comprehensive study of known QE and APE strategies, showing that a tighter coupling of the two is beneficial.
They show an improvement in BLEU and TER for English-Marathi, and to a lesser extent, also for English-German.

Hypothesis: QE can help APE
- sentence-level QE provides overall idea of how much editing is required
- word-level QE can help by identifying the tokens that need editing
The goal is to reduce over correction in APE, which is a common problem.

The main contributions are a joint training over QE and APE, which leads to improvements on two language pairs,
a novel multi-task strategy, and a comprehensive study of QE and APE combination strategies.



**Questions For The Authors:**

Question 1:
the EN-DE system does not benefit as much as the other one. Do you have ideas why?
(Might the dual-encoder setting also be beneficial for EN-EN, even though the languages share
script and some vocabulary? Is the initialization with IndicBERT optimal also for EN-DE? )

Question 2:
Data augmentation with phrase extraction:
I found this section a bit difficult to understand; if you extract phrases following the Moses routine,
I would expect the resulting phrases to be rather short, and maybe even not particularly helpful due
to not enough context (in contrast to the sentence-level triplets from the original data).
How many triplets do you get with this extraction step?

Question 3:
You state that your approach address over-correction (line 77 on page 1):
this seems straightforward to me in the QE as Activator and QE as Selector strategies, but it seems more complex in the joint variants.

Also, did you check whether the number of edits changed in the system variants?
Similarly, it would be interesting to know whether there are differences in error types that are addressed by the different system variants.


**Reasons To Accept:**

Previous research in QE and APE has mostly followed separate paths, and the authors combine both under the assumption that QE can help APE. The paper present a novels approach to multi-task learning; comparing different QE and APE strategies, the authors show that a tighter coupling is beneficial.
The results are consistent for both language pairs, even though EN-DE improved less.

**Reasons To Reject:**

The experiment is carried out on two language pairs, and it would be good to see whether the improvements are consistent for other language pairs as well (Assuming the availability of data sets...)
Also, it would be good to have Comet scores in addition to BLEU.

**Reproducibility:**

3: Could reproduce the results with some difficulty. The settings of parameters are underspecified or subjectively determined; the training/evaluation data are not widely available.

**Reviewer Confidence:**

2: Willing to defend my evaluation, but it is fairly likely that I missed some details, didn't understand some central points, or can't be sure about the novelty of the work.

---

> ### Author Rebuttal · Authors · 2023-08-28
>
> Thank you for reviewing our work and providing valuable comments and suggestions.
>
> ****
>
> ### The experiment is carried out on two language pairs:
>
> We conducted a comprehensive evaluation of the APE-QE combination methodologies using English-Marathi (En-Mr) and English-German (En-De) language pairs.
>
> These pairs offer distinct settings that encompass various factors:
> - Marathi and German languages belong to different language families.
> - English and German have a script and vocabulary overlap, while English and Marathi have different scripts and almost no vocabulary overlap.
> - Representing differing resource levels for Machine Translation (MT); where En-Mr stands as a low-resource language while En-De is recognized as a high-resource language.
> - The APE do-nothing baseline TER score is 22.93 for En-Mr, indicating relative ease, and 19.06 for En-De, signifying a more challenging scenario.
> - The available Synthetic and Real APE data comprises 2.5M and 18K triplets, respectively, for En-Mr, and 4M and 7K triplets, respectively, for En-De. This indicates a relative scarcity of synthetic data yet a significant amount of real APE data for En-Mr, in contrast to the data availability in En-De.
>
> **Given the diversity within these variations, our rationale was to ensure comprehensive coverage by evaluating the experiments across these two language pairs. As the evaluation of two language pairs doesn’t seem enough, we will conduct more experiments on the En-Ru pair and include the additional findings in the appendix.**
>
> ### Reporting Comet scores:
> We have reported TER and BLEU scores for the experiments, a convention in the APE community. However, we are happy to report the Comet scores in the paper.
>
> ### Question 1:
> **Low improvement for En-De**: The En-De APE Do-nothing baseline (19.06 TER points) is a bit tough, suggesting the presence of many high-quality translations in the test set. While the synthetic data is high in amount, the TER of synthetic data is higher than that of small-sized real APE data that contains 7K triplets.
>
> **Dual-encoder single-decoder for En-De**: Such architecture has been used before, and Correia et. al [1] showed that using a single-encoder single-decoder architecture achieves better performance for the En-De APE.
>
> **Use of IndicBERT for En-De**: IndicBERT-based encoder initialization may not be optimal for En-De. In the current work, our focus was to study whether we can improve APE with the help of QE using multitask-learning and not to achieve the best performance by focusing on other techniques and design of the architecture.
>
> ### Question 2:
> While the phrase-level triplets are not directly helpful in performing the sentence-level post-editing, we conjecture that they help the model learn the correction of phrases. It has been shown by the winning submission to the last year’s APE shared task. Only the synthetic APE data is augmented with the phrase-level APE triplets, not the real APE data. The number of augmented phrase-level APE triplets for En-Mr and En-De pairs are 50K and 60K, respectively. We will add this discussion to the paper to bring more clarity.
>
> ### Question 3, stats about translation edits, analysis based on error types:
> This observation is based on our qualitative analysis. We did not discuss it in more detail due to space limitations. We will add more qualitative examples with their analysis (including types of errors) and statistics on the number of translations that APE improved and deteriorated in the paper.
>
> ****
> **Reference:**
>
> [1] Gonçalo M. Correia and André F. T. Martins. 2019. A Simple and Effective Approach to Automatic Post-Editing with Transfer Learning. In Proceedings of the 57th Annual Meeting of the Association for Computational Linguistics, pages 3050–3056, Florence, Italy. Association for Computational Linguistics.

---

### Official Review · Reviewer_R7xZ · 2023-08-07

**Soundness:** 3

**Excitement:**

3: Ambivalent: It has merits (e.g., it reports state-of-the-art results, the idea is nice), but there are key weaknesses (e.g., it describes incremental work), and it can significantly benefit from another round of revision. However, I won't object to accepting it if my co-reviewers champion it.

**Missing References:**

Hokamp 2017 Ensembling Factored Neural Machine Translation Models for Automatic Post-Editing and Quality Estimation
Same model is used for APE and QE tasks

Marcin Junczys-Dowmunt and Roman Grundkiewicz. 2016. Log-linear combinations of monolingual and bilingual neural machine translation models for automatic post-editing.
Original synthetic data generation for APE paper



**Paper Topic And Main Contributions:**

The authors train a model with both APE and QE data in a multi-task learning (MTL) setting. The authors train a model with both Automatic Post-editing (APE) and quality estimation (QE) data in a multi-task learning (MTL) setting. The work explores four strategies to use QE to augment APE - (1) use QE as a gate to an APE model when the direct assessment (DA) score is below a threshold, (2) use QE to determine whether an APE output is better than the original, (3) provide QE outputs as additional inputs to APE, and (4) train a model using both QE and APE tasks in a MTL setting. On English-Marathi and EN-DE APE tasks, the jointly trained APE model shows some improvement over baselines.

**Questions For The Authors:**




**Reasons To Accept:**

- well written paper with good attention given to multi-task learning setup
- generally interesting multitask learning formulation for jointly training QE and APE

**Reasons To Reject:**

- low scientific novelty
- relatively small evaluation (only two language pairs)


**Reproducibility:**

4: Could mostly reproduce the results, but there may be some variation because of sample variance or minor variations in their interpretation of the protocol or method.

**Reviewer Confidence:**

4: Quite sure. I tried to check the important points carefully. It's unlikely, though conceivable, that I missed something that should affect my ratings.

---

> ### Author Rebuttal · Authors · 2023-08-28
>
> Thank you for reviewing our work and for your valuable comments and suggestions.
>
> ****
>
> ### Low Scientific Novelty:
>
> **Focus:** The focus of this work is to underline the benefit of using QE systems to improve APE systems when translations are obtained from an NMT system (in a black box setting).
>
> **Novelty:** While the application of multi-task learning in NLP is not new, we believe that the novelty of the work lies in introducing a previously unexplored direction: using multi-task learning to improve APE systems with the help of QE, especially in a jointly trained setting.
>
> **Implications:** We have demonstrated through comparison with the three existing QE-APE combination techniques that the multi-task-learning-based QE-APE combination technique outperforms them. We firmly believe that the implications of this work will not be limited to this immediate finding, but will also act as a foundational work for the APE community to explore this direction further.
>
> ### Relatively Small Evaluation (Only Two Language Pairs):
>
> We conducted a comprehensive evaluation of the APE-QE combination methodologies using the English-Marathi (En-Mr) and English-German (En-De) language pairs.
>
> These pairs offer distinct settings that encompass various factors:
> - Marathi and German languages belong to different language families.
> - English and German have a script and vocabulary overlap, while English and Marathi have different scripts, and almost no vocabulary overlap.
> - Representing differing resource levels for Machine Translation (MT); where En-Mr stands as a low-resource language pair while En-De is recognized as a high-resource language pair.
> - The APE do-nothing baseline TER score is 22.93 for En-Mr, indicating relative ease, and 19.06 for En-De, signifying a more challenging scenario.
> - The available Synthetic and Real APE data comprises 2.5M and 18K triplets, respectively, for En-Mr, and 4M and 7K triplets, respectively, for En-De. This indicates a relative scarcity of synthetic data yet a significant amount of real APE data for En-Mr, in contrast to the data availability in En-De.
>
> **Given the diversity within these variations, our rationale was to ensure comprehensive coverage by evaluating the experiments across these two language pairs. As the evaluation of two language pairs doesn’t seem enough, we will conduct more experiments on the En-Ru pair and include the additional findings in the appendix**.
>
> ### Missing References:
>
> Thank you for bringing it to our notice! We will cite these publications in our work.

---

### Meta-Review · Area_Chair_1eFx · 2023-09-20

**Recommendation:** 3

**Metareview:**

The authors investigate several methods to use a QE (sentence-level and word-level Quality Estimation) system to improve an APE (Automatic Post-Editing) system for machine translation.
They conclude a joint training of a model (dual-encoder single-decoder model with task-specific heads) with Nash-MTL (Mult Task Learning)
performs the best and obtains significant improvements in English-to-Marathi and English-to-German translation tasks.

The paper is well-written, and the experiments show the effectiveness of the proposed method.
However, one reviewer has judged that the proposed method has low scientific novelty because it merely applies a known MTL method to a slight extension of a standard APE model.

---

### Decision · Program_Chairs · 2023-10-07

**Decision:**

Accept-Findings

**Comment:**

The authors investigate several methods to use a QE (sentence-level and word-level Quality Estimation) system to improve an APE (Automatic Post-Editing) system for machine translation.
They conclude a joint training of a model (dual-encoder single-decoder model with task-specific heads) with Nash-MTL (Mult Task Learning)
performs the best and obtains significant improvements in English-to-Marathi and English-to-German translation tasks.

The paper is well-written, and the experiments show the effectiveness of the proposed method.
However, one reviewer has judged that the proposed method has low scientific novelty because it merely applies a known MTL method to a slight extension of a standard APE model.